# Effects of Long-Term Protein Restriction on Meat Quality and Muscle Metabolites of Shaziling Pigs

**DOI:** 10.3390/ani12152007

**Published:** 2022-08-08

**Authors:** Jie Zheng, Yehui Duan, Jiayi Yu, Fengna Li, Qiuping Guo, Tiejun Li, Yulong Yin

**Affiliations:** 1CAS Key Laboratory of Agro-Ecological Processes in Subtropical Region, Hunan Provincial Key Laboratory of Animal Nutritional Physiology and Metabolic Process, National Engineering Laboratory for Pollution Control and Waste Utilization in Livestock and Poultry Production, Institute of Subtropical Agriculture, Chinese Academy of Sciences, Changsha 410125, China; 2University of Chinese Academy of Sciences, Beijing 100039, China

**Keywords:** protein restriction, meat quality, muscle metabolites, Shaziling pigs

## Abstract

**Simple Summary:**

The effects of long-term protein restriction on meat quality of Shaziling pigs and the underlying mechanism remain relatively unexplored. The aim of this study is to investigate the changes in meat quality and muscle metabolites of Shaziling pigs under the condition of low-protein diets and to provide a practical nutritional manipulation for swine production. After a 24-week trial, we found that reducing dietary crude protein by 20% led to improved meat quality (the reduced L* value and the increased a* value) and altered metabolite profiles of longissimus thoracis, without impairing growth performance and carcass traits. Moreover, results showed that the improvement on meat quality might be credited to diminished concentrations of Danazol, N,N-dimethyl-Safingol, and cer(d18:0/14:0). Our findings suggested long-term protein restriction (20% reduction) is suitable to improve meat quality and sustain growth performance.

**Abstract:**

Background: It has been demonstrated that low-protein diets can improve the meat quality of pork. This study aimed to investigate the effects of long-term protein restriction from piglets to finishing pigs for 24 weeks on meat quality and muscle metabolites of Shaziling pigs. Results: Compared to the control group, reducing dietary protein levels by 20% reduced the L* value (*p* < 0.05), increased the a* value (*p* < 0.01), and tended to decrease pressing loss (*p* = 0.06) of *longissimus thoracis* muscle (LTM). Furthermore, compared to the control group, the −20% group had significantly lower levels of muscular danazol, N,N-dimethyl-Safingol, and cer(d18:0/14:0) (*p* < 0.05), all of which were positively associated with the L* value and negatively associated with the a* value (*p* < 0.05). Therefore, danazol, N,N-dimethyl-Safingol, and cer(d18:0/14:0) might be potential biomarkers for meat color. Conclusions: These results indicated that reducing dietary crude protein by 20% for 24 weeks could improve meat quality and alter muscular metabolites of Shaziling pigs, and the improvement in meat quality might be ascribable to decreased danazol, N,N-dimethyl-Safingol and cer(d18:0/14:0).

## 1. Introduction

Reducing the levels of dietary crude protein (CP) has been traditionally considered to favor the reduction in feed cost, nitrogen excretion, and the risk of gut disorders, without impairing the growth performance of pigs. Interestingly, besides these beneficial effects, accumulating evidence also shows that low-protein diets greatly increase the intramuscular fat content and improve meat quality [1,2,3]. In this sense, studies on the application of low-protein diets in animal production have been gaining attention. In the literature, information on the application of low-protein diets in non-Chinese commercial breeds is available. However, the effects of low-protein diets on the meat quality of native Chinese pig breeds are less well-known.

Non-Chinese commercial breeds and native Chinese pig breeds display distinctive differences in terms of meat quality, prolificacy, and growth. Specifically, native Chinese pig breeds have better meat quality (such as the cherry-red color, high levels of marbling, soft texture and superior flavor), lower growth rates, and lower lean meat rates when compared with non-Chinese commercial breeds [4,5,6]. With the improvement in people’s living standards, a growing demand for healthier, tastier, and more nutritious pork has emerged. In this context, native Chinese pig breeds are returning as focus of studies, mainly since they are a resource that could meet the diverse needs of consumers [7].

The Shaziling pig is a traditional fat-type native pig breed mainly reared in Hunan Province, China. It possesses excellent meat quality, strong disease resistance, and slow growth rates like most native Chinese pig breeds [8]. However, studies of Shaziling pigs are very limited, and previous studies have mainly focused on the comparison of carcass traits, meat quality, serum metabolome, and lipid metabolic and microbial profiles between Shaziling and Yorkshire pigs at different ages [9,10], and have discussed the differences in genetics, involving epigenetics, quantitative genetics, and metagenomics between Shaziling and Yorkshire pigs [8,11]. In view of the information above, our understanding of optimal nutrition interventions to regulate meat quality of Shaziling pigs is at its infancy, which hinders the breeding of Shaziling pigs and the promotion of meat products. Given the importance of nutrition in the quality control of animal meat, further research is required.

Metabolomics is a method that has the ability to detect subtle alterations in the content of low-molecular-weight metabolites in complex biological systems such as tissues and cells, resulting from external or internal factors. The past few decades have witnessed the development of metabolomics, and this emerging analytical platform shows valuable application in assessing meat quality. For example, ultra-high performance liquid chromatography–tandem mass spectrometry (UHPLC-MS/MS) aided in the investigation of serum potential biomarkers to predict meat quality of Shaziling pigs [10]. Non-targeted liquid chromatography–mass spectrometry has been used to characterize lipid-related metabolites in the muscle of pigs [12]. Xiao et al. adopted ^1^H nuclear magnetic resonance spectroscopy to analyze the metabolic composition of chicken meat [13]. Ultra-performance liquid chromatography–tandem mass spectrometry (UPLC-MS/MS) has been used to investigate the underlying mechanisms responsible for chlorogenic acid-induced improvements in meat quality of broiler breast muscles [14]. These studies highlighted the effectiveness of metabolomics for the evaluation of meat quality; hence, an approach integrating metabolomics and molecular biology should advance the understanding of the mechanism of meat quality changes in response to low-protein diets.

In this study, we determined the effects of long-term protein restriction for 24 weeks on meat quality of Shaziling pigs and showed muscle metabolite alterations resulting from low-protein diets via analysis of muscular metabolic profiling. 

## 2. Materials and Methods

### 2.1. Animals, Diets and Sample Collection

The experiment was approved by the Animal Care Committee of Institute of Subtropical Agriculture, Chinese Academy of Sciences and the ethic approval number is ISA-2020-023.

Forty Shaziling piglets (8.78 ± 0.33 kg, barrow) were randomly assigned to five dietary treatments (eight piglets per treatment): a control group in which piglets received a normal protein diet according to the nutrition requirement of Shaziling pigs (PDF S1), two high-protein groups, and two low-protein groups. The experiment lasted for 24 weeks, and separate diets were formulated for the following four periods: <15 kg, 15~30 kg, 30~50 kg, and >50 kg (Table 1 and Table 2). The dietary CP levels of the four periods in the control group were 18%, 16%, 14%, and 12%, respectively. The dietary CP levels of the four periods in the two high-protein groups were 20% and 10% higher than those in the control group, respectively. By contrast, the dietary CP levels of the four periods in the two low-protein groups were 20% and 10% lower than those in the control group, respectively. Therefore, in the following analysis, pigs of the five groups were abbreviated as +20%, +10%, 0 (control), −10%, and −20%, respectively.

Piglets were kept in individual cages with ad libitum access to food and water throughout the total trial period. Each pig was weighed at the beginning and at the termination of the four experimental periods for the calculation of average daily gain (ADG). Feed intake and feed refusals per pen were recorded weekly and at the end of the each experimental period for the calculation of average daily feed intake (ADFI) and the feed:gain ratio (F:G). All pigs were slaughtered at the end of the feeding test. After slaughter, the *longissimus thoracis* muscle (LTM) from the left side of the carcass was collected and refrigerated at 4 °C for meat quality data collection. Meanwhile, the LTM samples from the right side of the carcass was rapidly collected and stored at −80 °C for further analysis.

### 2.2. Carcass Traits

At slaughter, carcass weight was recorded after evisceration to calculate carcass yield. The distance from the anterior margin of the symphysis pubis to the fovea of the first cervical spine on the left half of the carcass was measured as carcass length (cm). Backfat thickness (mm) and loin eye area (height × width × 0.7 cm^2^) were measured between the 6th and 7th ribs using vernier caliper [5]. The lean/fat percentage was calculated as the percentage of lean/fat weight to carcass weight, respectively.

### 2.3. Meat Quality

Meat quality was reflected by measuring shear force, pH, color, and water-holding capacity (WHC). The shear force was measured by a Warner–Bratzler shear force device (TA. XT Plus, Stable Micro Systems, Godalming, UK) as previously described [14]. Meat pH values were assessed with a portable pH meter (Matthaus pH Star, Eckelsheim, Germany) at 45 min and 24 h post-mortem, respectively. Meat color traits (L*, lightness; a*, redness, and b*, yellowness) were determined by using a CR-410 hand-held chromameter (Kinica Minolta Sensing Inc., Osaka, Japan) at two different locations. The WHC parameters were tested using drip loss, pressing loss, and cooking loss, which were measured as we previously described [10,15,16]. 

### 2.4. Fatty Acid Composition and Health Lipid Indices

The fatty acid composition of LTM was determined via gas-liquid chromatography of methyl esters using an Agilent 7890A GC as previously described [17]. The concentration of every fatty acid was expressed as a percentage of total fatty acids. Next, we calculated the following health lipid indicies: the polyunsaturated fatty acids (PUFA): saturated fatty acids (SFA) ratio, n6:n3 PUFA ratio, atherogenicity (AI), thrombogenicity (TI), and peroxidisability (PI), desirable hypocholesterolemic fatty acids (DHFA), hypercholesterolemic saturated fatty acids (HSFA), and hypocholesterolemic and hypercholesterolemic fatty acids ratio (Hpo/Hper) [14,18].

### 2.5. UHPLC-MS/MS Analysis

Solid samples of 50 mg were accurately weighed, and the metabolites were extracted using a 400 µL methanol:water (4:1, *v*/*v*) solution. The mixture was treated by high throughput tissue crusher Wonbio-96c (Shanghai wanbo biotechnology Co., Ltd., Shanghai, China) at 50 Hz for 6 min, then followed by vortex for 30 s and ultrasound at 40 kHz for 30 min at 5 °C. The samples were placed at −20 °C for 30 min to precipitate proteins. After centrifugation at 13,000× *g* at 4 °C for 15 min, the supernatant was carefully transferred to sample vials for UHPLC-MS/MS analysis.

Chromatographic separation of the metabolites was performed on a Thermo UHPLC system equipped with an ACQUITY UPLC HSS T3 column (100 mm × 2.1 mm i.d., 1.8 µm; Waters, Milford, MA, USA). The mobile phases are as follows: 95% water and 5% acetonitrile (containing 0.1% formic acid) as solvent A; 47.5% acetonitrile, 47.5% isopropanol and 5% water (containing 0.1% formic acid) as solvent B. The solvent gradient changed according to Table 3. The sample injection volume was 2 μL, and the flow rate was set to 0.4 mL/min. The column temperature was maintained at 40 °C.

The mass spectrometric data were collected using a Thermo UHPLC-Q Exactive Mass Spectrometer equipped with an electrospray ionization (ESI) source operating in either positive or negative ion mode. The optimal conditions were set as follows: aux gas heater temperature, 425 °C; sheath gas flow rate 50 arb; aux gas flow rate 13 arb; ion-spray voltage floating (ISVF), −3500 V in negative mode and 3500 V in positive mode, respectively; normalized collision energy, 20–40–60 eV rolling for MS/MS. Data acquisition was performed with the data-dependent acquisition (DDA) mode. The detection was carried out over a mass range of 70–1050 *m*/*z*.

### 2.6. Statistical Analyses

Statistical analyses for the growth performance, carcass traits, meat quality, and fatty acid composition were performed by SAS 8.2 software (Institute, Inc., Cary, NC, USA) using the one-way ANOVA procedure, followed by Duncan’s multiple comparisons. When data did not conform to the normal distribution or homogeneity, the significance was conducted by the Kruskal–Wallis test. Results were presented as means ± standard error of the mean (SEM). *p* < 0.05 was considered significant, and 0.05 ≤ *p* < 0.10 was considered a trend. The metabonomics data were processed on the Majorbio Cloud Platform (https://cloud.majorbio.com, accessed on 13 June 2022). We used principal component analysis (PCA) and orthogonal partial least squares discriminant analysis (OPLS-DA) to differentiate metabolic profiles among groups. Differential metabolites were identified according to the standard of variable importance in the projection (VIP) > 1, *p* value < 0.05 in the OPLS-DA model. The correlational heatmaps of metabolites and meat quality were generated according to the result of Pearson correlation analysis.

## 3. Results

### 3.1. Growth Performance and Carcass Traits

As shown in Table 4, dietary treatments did not significantly affect the growth performance or carcass traits of Shaziling pigs. Moreover, neither linear nor quadratic effects were observed for these parameters (*p* > 0.05).

### 3.2. Meat Quality Traits

As presented in Table 5, the shear force value in the −10% group was significantly lower than that of the other four groups (*p* < 0.05), but no differences were detected among the four groups (*p* > 0.05). The L* value was highest in the control and −10% groups and lowest in the −20% group, with intermediate values in the other two groups (quadratic, *p* < 0.01). Compared with the control group, the a* value remained unchanged in the +10% group, and increased dramatically in the +20%, −10%, and −20% groups (quadratic, *p* < 0.05), although there was no significant difference among the three groups. Dietary treatments did not significantly affect the other parameters related to meat quality (*p* > 0.05).

### 3.3. Muscular Fatty Acid Composition and Health Lipid Indices

As revealed in Table 6, the contents of C14:0 and C17:0 decreased linearly in LTM of Shaziling pigs fed the decreasing level of protein in the diet, with the lowest values observed in the −20% group (*p* < 0.01). Compared to the control diet, the −10% protein diet markedly reduced the concentration of C20:4n6 (*p* < 0.05), whereas no significant difference was found in the −20% group. Moreover, decreasing dietary protein levels linearly decreased the concentrations of C18:3n3 and Σn3 PUFA (*p* < 0.01), but their concentrations did not differ between the control and the two low-protein groups. As shown in Table 7, the n6:n3 PUFA ratio (*p* < 0.01) and TI (*p* < 0.05) linearly increased as dietary protein levels decreased, but there was no significant difference between the control and −20% groups (*p* > 0.05). Dietary treatments did not significantly affect the ratio of PUFA to SFA, AI, PI, DHFA, HSFA, and the ratio of Hpo to Hper (*p* > 0.05).

### 3.4. Muscular Metabolomics Analyses

As shown in Appendix A, the shape of the peak is good, and the distribution is relatively uniform, suggesting the stability and reliability of the metabolomics profiles. Unlike unsupervised PCA, supervised OPLS-DA has the ability to detect the specific variables that lead to differences among groups [19]. Considering the reduced L* value and increased a* value in pigs offered low-protein diets (the −20% group vs. the control group), we performed OPLS-DA to explore the subtle differences in metabolic profiles among the two groups. As presented in Figure 1A, the scatter plots of the control group and the −20% group were well separated, suggesting a significant difference in the LTM metabolite patterns between the two groups.

Next, to show the alterations in the metabolite concentrations between the control and −20% group, a heat map was plotted. As revealed in Figure 2, the −20% group exhibited lower contents of danzol, N,N-dimethyl-Safingol, and cer(d18:0/14:0) (*p* < 0.05).

### 3.5. Correlation Analyses between Metabolites and Meat Quality Traits

As shown in Figure 3, the concentrations of danzol and cer(d18:0/14:0) were positively correlated with the L* value but negatively correlated with the a* value (*p* < 0.05). Similarly, the concentrations of N,N-dimethyl-Safingol had a positive correlation with the L* value (*p* < 0.05).

## 4. Discussion

The role of protein-restriction diets in saving protein resources and decreasing the emission of nitrogen in urea and feces has been well characterized. However, most of the studies related to protein restriction have focused on the short-term effects on non-Chinese commercial breeds [19]. Moreover, our understanding of the long-term effects of high-protein and low-protein diets on pig production performance (growth performance, carcass characteristics, and meat quality) is incomplete for native Chinese pig breeds. The long-term use of low-protein diet to save protein feed resources cannot be at the expense of pig production performance. The first factor to evaluate low-protein diets is growth performance, which includes ADG, ADFI, and F:G. Decades of studies have well-demonstrated that dietary CP reduction within 3% of the NRC (1998) did not impair the growth performance of growing–finishing pigs when supplemented with the first four limited amino acids (L-lysine, DL-methionine, L-threonine and L-tryptophan) [20]. However, an inhibitory effect on growth performance was observed when dietary CP levels were reduced by more than 3% with only the first four limited amino acids supplemented in diets [21]. Further studies have elucidated that compared with pigs fed high-protein diets, a reduction in CP by 4.8% along with the first four limited amino acids supplemented led to significantly reduced growth performance [22]. However, conflicting results were observed in the current study, which showed that reducing dietary CP level by 20% relative to the control group for 24 weeks resulted in similar growth performance in Shaziling pigs as those of control diets and long-term high-protein diets. These above findings suggest that long-term protein restriction (20% reduction in comparison to the control group) could be applied in Shaziling pigs without affecting their growth performance.

Carcass characteristics, including carcass length, carcass weight, dressing percentage, backfat thickness, loin eye area, lean percentage, and fat percentage, is the second factor to assess low-protein diets. Extensive past research has demonstrated that reducing dietary CP levels did not significantly affect dressing percentage [23,24,25]. These results are well-matched with our research since Shaziling pigs fed protein-restricted diets for 24 weeks had similar dressing percentage as that of control and high-protein diets. However, decreased loin eye area and increased backfat thickness at slaughter were consistently reported in pigs fed low-protein diets [26,27,28,29,30]. However, contradictory results are found when looking at long-term protein-restricted diets, which had no detrimental effect on loin eye area and backfat thickness of Shaziling pigs. This discrepancy might be related to response variation in different breeds (non-Chinese commercial breeds in previous studies vs. native Chinese pig breeds in this study), and these controversial data point out the necessity to further explore the mechanisms underlying these effects of long-term protein restriction on Shaziling pigs. These findings suggest a balance of energy to nitrogen in diets of the current study.

Meat quality, the third factor to evaluate low-protein diets, is primarily assessed by the following parameters: pH, color, WHC (drip loss, pressing loss, and cooking loss), and tenderness (shear force) [31]. Accumulating and emerging lines of evidence have revealed no significant influence of altering dietary CP on pH_24h_ and WHC of pigs [32,33]. In agreement with this, we found no significant difference in meat values of pH_45min_, pH_24h_, and WHC in Shaziling pigs fed different dietary CP. However, contradictory results are reported when looking at meat color. Several laboratory established that the values of L*, a*, and b* elevated when dietary CP was restricted [26,32]. In contrast, other studies demonstrated that altering dietary CP did not significantly affect the L* value [33,34]. The current study reports a significant reduction in L* value and an increase in a* value in pigs fed with low-protein diets (20% reduction in comparison to the control group) during the stage from 8.78 to 83.50 kg body weight. This finding suggests that low-protein diets lead to an improvement on meat quality of Shaziling pigs.

Fatty acid composition is an important contributor to various aspects of meat quality and nutritional value of meat and also the human health [35]. Studies have shown that SFA, such as C12:0, C14:0, C16:0, and C17:0, will raise the risk of cardiovascular disease and type 2 diabetes when over-consumed [36,37]. In this study, C12:0 and C17:0 greatly decreased when dietary protein was restricted by 20% relative to the control group. Therefore, the intake of Shaziling pigs fed protein-restricted diets (reducing dietary CP level by 20% relative to the control group) for 24 weeks may reduce the risk of cardiovascular disease. Conversely, unsaturated fatty acids, especially n3 PUFAs, are beneficial to human health and are also related to flavor and overall acceptability of meat [38]. In the current study, the n3 PUFA decreased with the low-protein diets but did not achieve statistical significance. These data suggest that reducing dietary CP level by 20% relative to the control group for 24 weeks did not significantly impair the flavor and overall acceptability of Shaziling pig meat. In addition, a PUFA:SFA ratio of above 0.4 has been recommended for meat [39], but in the present study, values were lower than this and were not significantly influenced by dietary CP levels. Moreover, higher values for Hpo/Hper and lower values for AI and TI are regarded to be healthier [18]. Our current study showed that these parameters were not significantly different between the control and the −20% group. Overall, from the perspective of fatty acid composition, reducing dietary CP level by 20% relative to the control group for 24 weeks could produce healthy pork as the control group.

Metabolomics is applied for the investigation of key metabolites contributing to the physico-chemical properties, and hence, it helps to account for meat quality traits [40]. In the current study, a comparative analysis of muscular metabolome was performed between the control and −20% group, since the −20% group improved meat quality without impairing growth performance and carcass characteristics. We found that the −20% group exhibited significantly lower concentrations of Danazol, N,N-dimethyl-Safingol, and cer(d18:0/14:0) compared with the control group. Moreover, the correlation analysis showed that the three metabolites were negatively associated with the a* value and positively related to the L* value. Therefore, based on these data and the abovementioned data concerning meat quality, it is postulated that Danazol, N,N-dimethyl-Safingol, and cer(d18:0/14:0) might be potential biomarkers of the −20% group and might be implicated in various pathways for improved meat quality upon protein-restricted diets.

In conclusion, the current study suggested that the long-term ingestion of a protein-restricted diet could improve the meat quality of Shaziling pigs without impairing their growth performance and carcass characteristics. Furthermore, long-term protein restriction reduced the metabolites (including Danazol, N,N-dimethyl-Safingol, and cer(d18:0/14:0)) in the muscle of Shaziling pigs, which may help to explain the improvement in meat quality. Taken together, the above findings provide a molecular basis for designing nutritional and effective feeding strategy for Shaziling pigs to improve meat quality and sustain their growth performance.

## Figures and Tables

**Figure 1 animals-12-02007-f001:**
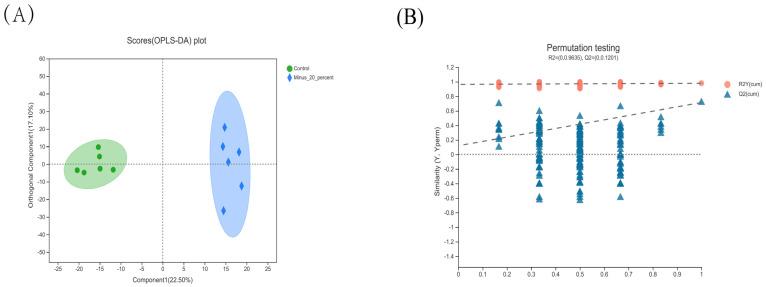
OPLS-DA score plot and OPLS-DA model validation diagram for metabolomics data in *longissimus thoracis* muscle of Shaziling pigs (*n* = 6). (**A**) OPLS-DA score plot of LTM metabolites (control vs. −20%). (**B**) OPLS-DA model validation diagram.

**Figure 2 animals-12-02007-f002:**
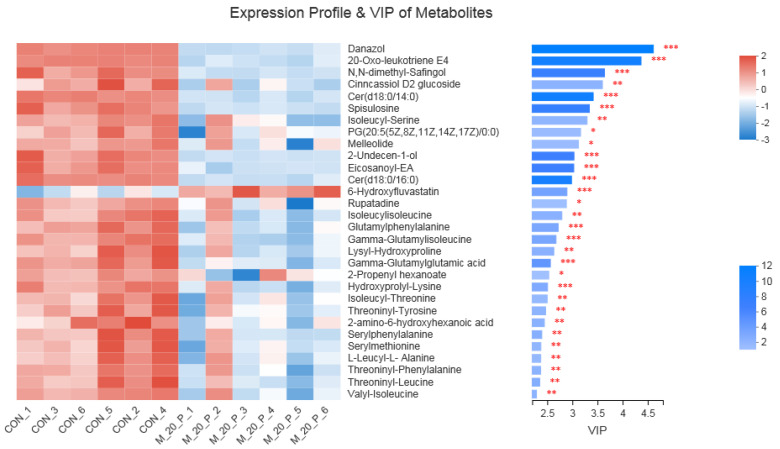
Heatmap of top 30 differential metabolites in *longissimus thoracis* muscle between the control group and −20% group (*n* = 6). CON = control, M_20_*p* = −20%. Significance levels were marked as * for *p* < 0.05, ** for *p* < 0.01, and *** for *p* < 0.001.

**Figure 3 animals-12-02007-f003:**
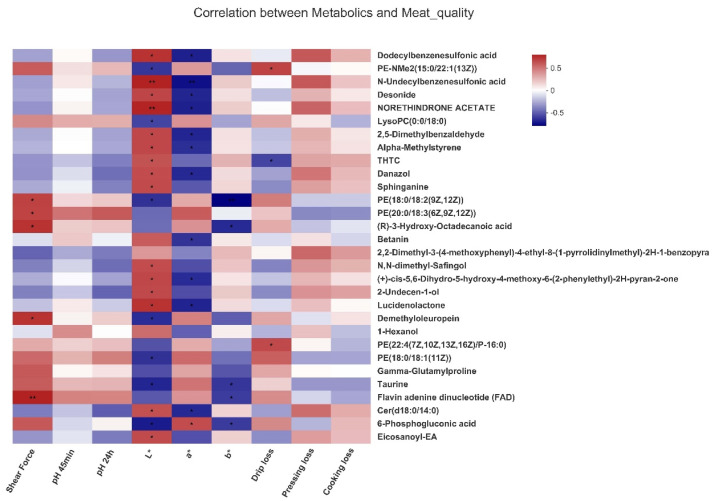
Correlations between *longissimus thoracis* muscle metabolites and meat quality traits (−20% group, *n* = 6). The top 30 metabolites were selected. Significance levels were marked as * for *p* < 0.05, ** for *p* < 0.01.

**Table 1 animals-12-02007-t001:** Ingredients and nutrient levels of the experimental diets for Shaziling pigs (<15 kg and 15~30 kg) (air-dried, %).

	Dietary Protein Levels									
Item	<15 kg					15–30 kg				
	+20%	+10%	0	−10%	−20%	+20%	+10%	0	−10%	−20%
Ingredients, %										
Corn	50.38	53.12	59.12	62.48	68.3	48.52	53.63	58.7	64.6	68.5
Soybean meal	26.95	27.31	22.21	20.5	15.5	29.31	24.2	18	12.3	6
Corn gluten meal	9	5	3.74	1.5	0	0	0	0.2	0.1	0.15
Wheat bran	9	9	9	9	9	18	18	19.1	18.8	20.9
Soybean oil	0.7	1.4	1.4	1.8	2.2	0	0	0	0	0
Lys	0.35	0.33	0.55	0.65	0.75	0.33	0.33	0.43	0.55	0.7
Met	0.18	0.3	0.3	0.32	0.35	0.3	0.3	0.3	0.3	0.3
Thr	0.1	0.2	0.26	0.3	0.4	0.2	0.2	0.3	0.3	0.4
Trp	0.02	0.02	0.05	0.08	0.08	0.02	0.02	0.05	0.15	0.15
CaHPO_4_	1	1	1.05	1.05	1.1	1	1	0.8	0.8	0.8
CaCO_3_	1.02	1.02	1.02	1.02	1.02	1.02	1.02	0.82	0.8	0.8
Salt	0.3	0.3	0.3	0.3	0.3	0.3	0.3	0.3	0.3	0.3
Premix ^a^	1	1	1	1	1	1	1	1	1	1
Nutrient levels ^b^										
Digestible energy, MJ/kg	13.5	13.54	13.48	13.49	13.5	12.78	12.78	12.76	12.74	12.6
Crude protein, %	21.6	19.94	18	16.54	14.41	19.24	17.62	16.11	14.43	12.92
Calcium, %	0.72	0.72	0.71	0.71	0.7	0.73	0.72	0.58	0.55	0.53
Total phosphorus, %	0.59	0.58	0.57	0.56	0.55	0.65	0.63	0.58	0.56	0.55
Available phosphorus, %	0.27	0.26	0.27	0.26	0.26	0.29	0.28	0.25	0.24	0.24

^a^ Premix provided for 1 kg of the diet: Cu, 128 mg; Mn, 97.6 mg; Zn 109 mg; Fe, 197.6 mg; Se, 1 mg; I, 1 mg; Co, 1 mg; VA, 32,500 IU; VD3, 10,000 IU; VE, 80 IU; VK3, 10 mg; VB1, 10 mg/kg; VB2, 25 mg; VB6, 8 mg; VB12, 0.075 mg; biotin, 0.075 mg; folic acid, 5 mg; nicotinamide, 100 mg; pantothenic acid, 50 mg; choline, 1600 mg; mildewcide, 0.10%; ethoxyquinoline (33%), 0.05%; acidifier, 0.25%. ^b^ Crude protein was measured, and others were calculated.

**Table 2 animals-12-02007-t002:** Ingredients and nutrient levels of the experimental diets for Shaziling pigs (30~50 kg and >50 kg) (air-dried, %).

	Dietary Protein Levels									
Item	30–50 kg					>50 kg				
	+20%	+10%	0	−10%	−20%	+20%	+10%	0	−10%	−20%
Ingredients, %										
Corn	47.32	51.76	56.07	61.58	67.04	62.04	66.04	70.55	74.35	78.63
Soybean meal	17.72	13	8	3.3	0	14	10	5.5	1.7	0
Corn gluten meal	0	0	0	0	0	0	0	0	0	0
Wheat bran	31.18	31.46	31.96	30.96	28.5	20.18	20.18	20.18	20.18	16
Soybean oil	0	0	0	0	0	0	0	0	0	0
Lys	0.32	0.32	0.45	0.55	0.6	0.32	0.32	0.45	0.36	0.4
Met	0.2	0.2	0.2	0.22	0.22	0.2	0.2	0.2	0.2	0.2
Thr	0.12	0.16	0	0.25	0.27	0.12	0.16	0.2	0.25	0.2
Trp	0.02	0.05	0.07	0.09	0.11	0.02	0.05	0.07	0.09	0.1
CaHPO_4_	0.8	0.73	0.73	0.73	0.76	0.8	0.73	0.73	0.73	1.61
CaCO_3_	1.02	1.02	1.02	1.02	1.2	1.02	1.02	0.82	0.84	1.56
Salt	0.3	0.3	0.3	0.3	0.3	0.3	0.3	0.3	0.3	0.3
Premix ^a^	1	1	1	1	1	1	1	1	1	1
Nutrient levels ^b^										
Digestible energy, MJ/kg	12.19	12.18	12.12	12.74	12.23	12.73	12.73	12.74	12.74	12.71
Crude protein, %	16.81	15.41	14.06	14.43	11.4	14.46	13.27	12.03	10.81	9.71
Calcium, %	0.67	0.64	0.62	0.55	0.67	0.65	0.62	0.53	0.52	0.98
Total phosphorus, %	0.66	0.63	0.62	0.56	0.57	0.57	0.55	0.53	0.52	0.63
Available phosphorus, %	0.28	0.27	0.26	0.24	0.25	0.25	0.23	0.23	0.22	0.34

^a^ Premix provided for 1 kg of the diet: Cu, 128 mg; Mn, 97.6 mg; Zn 109 mg; Fe, 197.6 mg; Se, 1 mg; I, 1 mg; Co, 1 mg; VA, 32,500 IU; VD3, 10,000 IU; VE, 80 IU; VK3, 10 mg; VB1, 10 mg/kg; VB2, 25 mg; VB6, 8 mg; VB12, 0.075 mg; biotin, 0.075 mg; folic acid, 5 mg; nicotinamide, 100 mg; pantothenic acid, 50 mg; choline, 1600 mg; mildewcide, 0.10%; ethoxyquinoline (33%), 0.05%; acidifier, 0.25%. ^b^ Crude protein was measured, and others were calculated.

**Table 3 animals-12-02007-t003:** The elution gradient of the mobile phase.

Time (min)	Flow Rate (mL/min)	A (%)	B (%)
0	0.4	100	0
3.5	0.4	75.5	24.5
5	0.4	35	65
5.5	0.4	0	100
7.4	0.6	0	100
7.6	0.6	48.5	51.5
7.8	0.5	100	0
9	0.4	100	0
10	0.4	100	0

**Table 4 animals-12-02007-t004:** Effects of long-term protein restriction on growth performance and carcass traits of Shaziling pigs.

Item	Dietary Levels of Protein					SEM	*p*-Value		
	+20%	+10%	0	−10%	−20%		ANOVA	Linear	Quadratic
Initial weight, kg	8.74	8.84	8.99	8.54	8.78	0.33	1.00	0.93	0.99
Final weight, kg	84.95	80.49	84.28	85.83	81.96	1.87	0.90	0.96	1.00
ADFI, kg/d ^1^	1.74	1.76	1.72	1.78	1.70	0.04	0.97	0.87	0.95
ADG, kg/d ^2^	0.41	0.39	0.40	0.42	0.40	0.01	0.78	0.97	0.99
F:G ^3^	4.19	4.25	4.28	4.24	3.92	0.09	0.68	0.37	0.34
Carcass length, cm	81.66	80.24	79.41	80.45	79.25	0.56	0.68	0.24	0.45
Carcass weight, kg	46.29	45.94	45.99	48.2	46.24	1.08	0.94	0.80	0.96
Dressing percentage, %	54.45	54.29	55.33	56.23	56.04	<0.01	0.59	0.12	0.30
Backfat thickness, mm	22.98	20.32	25.64	26.84	20.99	0.80	0.11	0.75	0.29
Loin eye area, cm^2^	10.97	12.02	13.08	12.25	11.46	0.55	0.81	0.80	0.47
Lean percentage, %	38.32	38.59	37.57	35.39	40.62	<0.01	0.19	0.77	0.31
Fat percentage, %	35.90	34.45	37.92	38.36	35.56	<0.01	0.50	0.59	0.59

^1^ ADFI = Average daily feed intake. ^2^ ADG = Average daily gain. ^3^ F:G = Feed:gain.

**Table 5 animals-12-02007-t005:** Effects of long-term protein restriction on meat quality traits in *longissimus thoracis* muscle of Shaziling pigs.

Item	Dietary Levels of Protein					SEM	*p*-Value		
	+20%	+10%	0	−10%	−20%		ANOVA	Linear	Quadratic
Shear force, N	44.04 ^a^	42.23 ^a^	44.31 ^a^	33.26 ^b^	51.50 ^a^	1.90	0.01	0.66	0.18
pH_45min_	6.50	6.55	6.58	6.62	6.55	0.04	0.95	0.58	0.72
pH_24h_	5.63	5.68	5.64	5.65	5.66	0.01	0.67	0.56	0.79
L*-lightness	44.66 ^bc^	46.13 ^ab^	46.15 ^a^	46.36 ^a^	44.29^c^	0.27	0.02	0.94	<0.01
a*-redness	17.36 ^A^	16.28 ^BC^	15.60 ^C^	16.88 ^AB^	16.83 ^AB^	0.17	<0.01	0.75	0.02
b*-yellowness	5.99	6.14	5.77	6.28	5.53	0.13	0.36	0.39	0.49
Drip loss, %	2.68	2.72	2.55	2.23	3.30	<0.01	0.14	0.37	0.15
Pressing loss, %	32.17	27.66	31.14	28.38	27.91	<0.01	0.06	0.06	0.16
Cooking loss, %	17.13	15.86	15.49	14.76	15.25	<0.01	0.63	0.16	0.28

^A,B,C^ Values (*n* = 6–8) within a row with different capital superscripts differ significantly (*p* < 0.05) by Duncan’s multiple comparisons. ^a,b,c^ Values (*n* = 6–8) within a row with different lowercase differ significantly (*p* < 0.05) by the Kruskal–Wallis test.

**Table 6 animals-12-02007-t006:** Effects of long-term protein restriction on fatty acid composition in *longissimus thoracis* muscle of Shaziling pigs.

Item	Dietary Protein Levels					SEM	*p*-Value		
	+20%	10%	0	−10%	−20%		ANOVA	Linear	Quadratic
C10:0	0.09	0.09	0.09	0.09	0.08	0.00002	0.25	0.17	0.08
C12:0	0.07 ^A^	0.08 ^A^	0.08 ^A^	0.07 ^AB^	0.06 ^B^	0.00002	0.01	0.06	<0.01
C14:0	1.51 ^a^	1.31 ^bc^	1.35 ^bc^	1.41 ^b^	1.23 ^c^	0.00026	<0.01	<0.01	0.03
C16:0	28.84	28.36	28.71	28.82	28.16	0.00152	0.68	0.42	0.65
C16:1	3.12	2.88	2.68	3.01	2.79	0.00072	0.06	0.28	0.37
C17:0	0.17 ^A^	0.17 ^A^	0.17 ^A^	0.16 ^AB^	0.15 ^B^	0.00003	0.02	<0.01	<0.01
C18:0	15.73	16.08	17.37	16.67	16.75	0.00193	0.08	<0.05	0.04
C18:1n9t	0.14 ^A^	0.14 ^AB^	0.12 ^B^	0.14 ^AB^	0.13 ^AB^	0.00002	<0.05	0.05	0.06
C18:1n9c	40.66	38.87	38.37	39.87	39.07	0.00452	0.54	0.48	0.46
C18:2n6c	6.41	7.92	7.57	6.64	7.5	0.00317	0.55	0.67	0.72
C20:0	0.27	0.27	0.29	0.25	0.29	0.00007	0.38	0.68	0.84
C20:1	0.82	0.85	0.76	0.82	0.73	0.00017	0.19	0.10	0.21
C18:3n3	0.23 ^A^	0.23 ^A^	0.20 ^AB^	0.20 ^AB^	0.17 ^B^	0.00006	<0.01	<0.01	<0.01
C20:2	0.23	0.26	0.23	0.22	0.23	0.00007	0.29	0.40	0.55
C20:3n6	0.19	0.22	0.28	0.21	0.24	0.00013	0.27	0.44	0.39
C20:4n6	1.31 ^b^	2.21 ^ab^	2.29 ^a^	1.48 ^b^	2.02 ^ab^	0.00143	<0.05	0.48	0.33
SFA ^1^	46.69	46.36	48.06	47.48	46.72	0.00216	0.23	0.31	0.09
MUFA ^2^	44.74	42.74	41.94	43.84	42.72	0.00505	0.34	0.24	0.20
PUFA ^3^	8.37	10.85	10.57	8.74	10.15	0.00469	0.39	0.56	0.47
Σn6 PUFA ^4^	7.91	10.35	10.14	8.33	9.75	0.00462	0.33	0.52	0.46
Σn3 PUFA ^5^	0.23 ^A^	0.23 ^A^	0.20 ^AB^	0.20 ^AB^	0.17 ^B^	0.00006	<0.01	<0.01	<0.01

SFA ^1^ = C10:0 + C12:0 + C14:0 + C16:0 + C17:0 + C18:0 + C20:0. MUFA ^2^ = C16:1 + C18:1n9t + C18:1n9c + C20:1 + C22:1n9. PUFA ^3^ = C18:2n6c + C18:3n3 + C20:2 + C20:3n6 + C20:4n6. n6 PUFA ^4^ = C18:2n6c + C20:3n6 + C20:4n6. n3 PUFA ^5^ = C18:3n3. ^A,B^ Values (*n* = 6–8) within a row with different capital superscripts differ significantly (*p* < 0.05) by Duncan’s multiple comparisons. ^a,b,c^ Values (*n* = 6–8) within a row with different lowercase differ significantly (*p* < 0.05) by the Kruskal–Wallis test.

**Table 7 animals-12-02007-t007:** Effects of long-term protein restriction on health lipid indices in *longissimus thoracis* muscle of Shaziling pigs.

Item	Dietary Protein Levels					SEM	*p*-Value		
	+20%	10%	0	−10%	−20%		ANOVA	Linear	Quadratic
n6:n3 PUFA	34.79 ^b^	44.57 ^ab^	51.92 ^a^	43.26 ^ab^	56.13 ^a^	0.023	0.04	<0.01	0.03
ΣPUFA:SFA	0.17	0.21	0.22	0.18	0.21	0.010	0.29	0.34	0.44
AI	0.65	0.63	0.65	0.67	0.63	0.007	0.36	0.98	0.69
TI	1.52 ^b^	1.54 ^ab^	1.64 ^a^	1.62 ^a^	1.63 ^a^	0.018	0.04	0.01	0.03
PI	0.12	0.20	0.18	0.15	0.17	0.010	0.09	0.72	0.27
DHFA	0.69	0.70	0.69	0.69	0.70	0.002	0.31	0.28	0.52
HSFA	0.30	0.30	0.30	0.30	0.29	0.002	0.48	0.65	0.66
Hpo/Hper	1.59	1.59	1.54	1.53	1.60	0.013	0.43	0.65	0.33

^a,b^ Values (*n* = 6–7) within a row with different lowercase differ significantly (*p* < 0.05) by the Kruskal–Wallis test.

## Data Availability

The data presented in this study are available on request from the corresponding author.

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
