# Peer review of "Effects of Long-Term Protein Restriction on Meat Quality and Muscle Metabolites of Shaziling Pigs"

_animals, 2022, doi:10.3390/ani12152007_

Round 1

Reviewer 1 Report

1. In Introduction section at the first sentence CP (crude protein) should be added because its first use.

2. In Introduction section at the second paragraph “Chinese indigenous pigs have better meat quality” in terms of what?

3. in Section 2,6, Statistical Analysis “ANOVA procedure, followed by Duncan’s multiple comparisons or Kruskal–Wallis test.” Kruskal–Wallis is not a posterior test of ANOVA. If Kruskal–Wallis test used for non-normal data analysis please write it clearly and mention which statistic used for binary combinations.

4. in Table 3 “a,bValues (n = 6-8) within a row with different capital superscripts differ significantly (P < 0.05) by Duncan’s multiple comparisons. A,B,CValues (n = 6-8) within a row with different lowercase differ significantly (P < 0.05) by the Kruskal–Wallis test.” There is no letters in the table.

5. For Figure 1 (A) total explanation rate of PCA is 24.4% that is too low.

Author Response

Dear Editors:

Thanks for the constructive suggestions and comments from Editorial Boarding and reviewers. We have read the referees' comments very carefully, have consulted and discussed the reviewers' comments with several professors, and now we have further revised the comments according to the reviewers' suggestions to improve the manuscript. At the same time, we indexed revisions in red color in the manuscript. Response to the comments of 1849865 was listed following with ‘A’ for answers.

Thank you very much for your considering our manuscript. We are looking forward to hearing from you soon.

Best regards.

Yours,

Yehui Duan

Reviewer #1

  1. In Introduction section at the first sentence CP (crude protein) should be added because its first use.

A: Thanks for the reviewer’s suggestion. We have added “CP (crude protein)” in the first sentence.   

  1. In Introduction section at the second paragraph “Chinese indigenous pigs have better meat quality” in terms of what?

A: Enjoyable sensory attributes, such as the cherry-red color, high levels of marbling, soft texture and superior flavour.   

  1. in Section 2,6, Statistical Analysis “ANOVA procedure, followed by Duncan’s multiple comparisons or Kruskal–Wallis test.” Kruskal–Wallis is not a posterior test of ANOVA. If Kruskal–Wallis test used for non-normal data analysis please write it clearly and mention which statistic used for binary combinations.

A: Thanks for the reviewer’s suggestion. When data did not conform to the normal distribution or homogeneity, the significance was conducted by the Kruskal–Wallis test. A,B,CValues (n = 6-8) within a row with different capital superscripts differ significantly (P < 0.05) by Duncan’s multiple comparisons. a,b,cValues (n = 6-8) within a row with different lowercase differ significantly (P < 0.05) by the Kruskal–Wallis test. We have added these descriptions in Section 2,6, Statistical Analysis and footnotes.

  1. in Table 3 “a,bValues (n = 6-8) within a row with different capital superscripts differ significantly (P < 0.05) by Duncan’s multiple comparisons. A,B,CValues (n = 6-8) within a row with different lowercase differ significantly (P < 0.05) by the Kruskal–Wallis test.” There is no letters in the table.

A: We have improved the footnote according to your suggestion.   

  1. For Figure 1 (A) total explanation rate of PCA is 24.4% that is too low.

A: Thanks for the reviewer’s suggestion. Unlike unsupervised PCA, supervised OPLS-DA has the ability to detect the specific variables that leads to differences among groups. Considering the reduced L* value and increased a* value in pigs offered low-protein diets (the -20% group vs the control group), we performed OPLS-DA to explore the subtle differences in metabolic profiles among the two groups. As presented in Fig. 1A, the scatter plots of the control group and the -20% group were well separated, suggesting a significant difference in the LTM metabolite patterns be-tween the two groups. 

Reviewer 2 Report

Dear editor 

The aim of Manuscript ID animals-1849865, entitled “Effects of long-term protein restriction on meat quality and muscle metabolites of Shaziling pigs’’ deals with an interesting topic and fits with the scope of the journal. 

The current study provides new data regarding the effects of long-term protein restriction for 24 weeks on meat quality of native Chinese pig breeds.

Generally, the current article is a well-designed, providing novel results. However, an English editing is required in some parts of this article.

Therefore, I suggest that the article be accepted for publication, under minor revision.

Please let me congratulate you on the quality of your journal and thank you for giving me the opportunity to contribute as a reviewer.

Comments to the authors          

Introduction 

§  dietary CP = dietary crude protein (CP)

§  Chinese indigenous pigs= native Chinese pig breeds

§  Foreign hybrid pigs = non- Chinese commercial breeds  

Materials and Methods

§  Please provide Ethical Note for your study, including the approval number by the Ethics Committee 

§  Correct the font size in Table 1

§  Carcass Traits = add appropriate reference

Discussion

§  Chinese local pigs = native Chinese pig breeds

§  These findings suggest that long-term protein restriction (20% reduction relative to the control group for 24 weeks) would be possible for Shaziling pigs without affecting pig growth performance = These above findings suggest that long-term protein restriction (20% reduction in comparison to the control group) could be applied in Shaziling pigs, without affecting their growth performance.

§  The current study suggested a reduction of L* value and an increase of a* value in pigs fed with low-protein diets (20% reduction relative to the control group) from 8.78 to 83.50 kg body weight, suggesting an improved meat quality = The current study reports a significant  reduction of L* value and an increase of a* value in pigs fed with low-protein diets (20% reduction in comparison to the control group) during the stage from 8.78 to 83.50 kg body weight. This finding suggest that low-protein diets lead to an improvement on meat quality of Shaziling pigs

§  Taken together, these findings provide a molecular basis for designing nutritional and effective strategy for Shaziling pigs to improve meat quality while sustain growth traits = the above findings provide a molecular basis for designing nutritional and effective feeding strategy for Shaziling pigs to improve meat quality and sustain their growth performance

Author Response

Dear Editors:

Thanks for the constructive suggestions and comments from Editorial Boarding and reviewers. We have read the referees' comments very carefully, have consulted and discussed the reviewers' comments with several professors, and now we have further revised the comments according to the reviewers' suggestions to improve the manuscript. At the same time, we indexed revisions in red color in the manuscript. Response to the comments of 1849865 was listed following with ‘A’ for answers.

Thank you very much for your considering our manuscript. We are looking forward to hearing from you soon.

Best regards.

Yours,

Yehui Duan

Reviewer #2

Thanks a lot for the reviewer’s suggestions. We have revised the manuscript in the light of your report.

  1. dietary CP = dietary crude protein (CP)

A: Thanks for the reviewer’s suggestion. We have added “CP (crude protein)” in the first sentence.

  1. Chinese indigenous pigs= native Chinese pig breeds

A: We have changed “Chinese indigenous pigs” to “native Chinese pig breeds”.

  1. Foreign hybrid pigs = non- Chinese commercial breeds.

A: We have changed “Foreign hybrid pigs” to “non- Chinese commercial breeds”.

  1. Please provide Ethical Note for your study, including the approval number by the Ethics Committee.

A: We have provided Ethical Note and the approval number by the Ethics Committee in materials and methods.

  1. Correct the font size in Table 1

A: We have corrected the font size in Table 1.

  1. Carcass Traits = add appropriate reference

A: We have cited a study named “Comparisons of carcass traits, meat quality, and serum metabolome between Shaziling and Yorkshire pigs”.

  1. Chinese local pigs = native Chinese pig breeds

A: We have changed “Chinese local pigs” to “native Chinese pig breeds”.

  1. These findings suggest that long-term protein restriction (20% reduction relative to the control group for 24 weeks) would be possible for Shaziling pigs without affecting pig growth performance = These above findings suggest that long-term protein restriction (20% reduction in comparison to the control group) could be applied in Shaziling pigs, without affecting their growth performance.

A: We have changed “These findings suggest that long-term protein restriction (20% reduction relative to the control group for 24 weeks) would be possible for Shaziling pigs without affecting pig growth performance” to “These above findings suggest that long-term protein restriction (20% reduction in comparison to the control group) could be applied in Shaziling pigs, without affecting their growth performance”.

  1. The current study suggested a reduction of L* value and an increase of a* value in pigs fed with low-protein diets (20% reduction relative to the control group) from 8.78 to 83.50 kg body weight, suggesting an improved meat quality = The current study reports a significant reduction of L* value and an increase of a* value in pigs fed with low-protein diets (20% reduction in comparison to the control group) during the stage from 8.78 to 83.50 kg body weight. This finding suggest that low-protein diets lead to an improvement on meat quality of Shaziling pigs

A: We have changed “The current study suggested a reduction of L* value and an increase of a* value in pigs fed with low-protein diets (20% reduction relative to the control group) from 8.78 to 83.50 kg body weight, suggesting an improved meat quality” to “The current study reports a significant reduction of L* value and an increase of a* value in pigs fed with low-protein diets (20% reduction in comparison to the control group) during the stage from 8.78 to 83.50 kg body weight. This finding suggest that low-protein diets lead to an improvement on meat quality of Shaziling pigs”.

  1. Taken together, these findings provide a molecular basis for designing nutritional and effective strategy for Shaziling pigs to improve meat quality while sustain growth traits = the above findings provide a molecular basis for designing nutritional and effective feeding strategy for Shaziling pigs to improve meat quality and sustain their growth performance

A: We have changed “Taken together, these findings provide a molecular basis for designing nutritional and effective strategy for Shaziling pigs to improve meat quality while sustain growth traits” to “the above findings provide a molecular basis for designing nutritional and effective feeding strategy for Shaziling pigs to improve meat quality and sustain their growth performance”

Round 2

Reviewer 1 Report

The authors made changes according to the comments. This manuscript can be published as it.